# (How) Can AI Bots Lie?[*]

## *A Formal Perspective on Explanations, Lies, and the Art of Persuasion*

**Tathagata Chakraborti**[1]  and  **Subbarao Kambhampati**[2]

[1]IBM Research AI
[2]Arizona State University

tchakra2@ibm.com, rao@asu.edu

## Abstract

Recent work on explanations [Chakraborti *et al.*, 2017] for decision-making problems has viewed the explanation process as one of *model reconciliation* where an AI agent brings the human mental model (of its capabilities, beliefs, and goals) to the same page with regards to a task at hand. This formulation succinctly captures many possible types of explanations, as well as explicitly addresses the various properties – e.g. the social aspects, contrastiveness, and selectiveness – of explanations [Miller, 2018] studied in social sciences among human-human interactions. However, it turns out that the same process can be hijacked into producing "alternative explanations" that are not true but still satisfy all these properties of a proper explanation. In AIES 2019, we discussed when such behavior may be appropriate but did not go into details of how exactly they can be generated. In this paper, we go into details of this curious feature of the model reconciliation process as a well established framework for explanation generation of decision making problems and formalize the relationship between explanations, lies, and persuasion in the model reconciliation framework.

## 1 The Model Reconciliation Process

One of the root causes for the need of an explanation is that of model differences between the human and the AI agent. This is because, even if an agent makes the best decisions possible given its model, they may appear to be suboptimal or *inexplicable* if the human has a different mental model of its capabilities, beliefs and goals. Thus, it follows that the explanation process, whereby the AI agent justifies its behavior to the human in the loop, is one of *model reconciliation*. This approach to explainable AI was formalized recently in [Chakraborti *et al.*, 2017] where the authors posit that an explanation process cannot be a soliloquy but instead address the mental model of the explainee. This follows from similar insights from investigation done in the social sciences, as

explored in detail in [Miller, 2018]. In short, the model reconciliation process ensures that after an explanation, there are no better foils than the given decision in the updated mental model of the explainee – i.e. the human would agree that the decision made by the agent was the best possible.

In the model reconciliation framework, the mental model is a version of the decision making problem at hand which the agent believes the human is operating under. The actual decision making problem may be over a graph, a planning problem, a logic program, etc. The concepts discussed in the paper are agnostic of the actual representation.

**The Model Reconciliation Process** $\langle M^R, M_h^R, \pi \rangle$ takes in the agent model $M^R$, the human mental model of it $M_h^R$, and the agent decision $\pi$ which is optimal[1] in $M^R$ as inputs and produces a model $\bar{M}_h^R$ where $\pi$ is also optimal.

**An Explanation** $\epsilon$ is the model difference $\bar{M}_h^R \Delta M_h^R$.

Thus, by setting the mental model $\bar{M}_h^R \leftarrow M_h^R + \epsilon$ (by means of some form of interaction / communication), the human cannot come up with a better *foil* or decision $\hat{\pi}$, and hence we say that the original decision $\pi$ has been *explained*. This is referred to as the **contrastive property** of an explanation.[2] This property is also the basis of persuasion since the human, given this information, cannot come up with any other alternative to what was done.

So how do we compute this model update? It turns out that there are several possibilities [Chakraborti *et al.*, 2017], many of which have the contrastive property.

**Minimal Explanations**
These minimize the size of an explanation and ensure that the human cannot find a better foil using the fewest number of model updates. These are referred to as *minimally complete explanations* or MCEs.

$$\epsilon_{MCE} = \arg\min \bar{M}_h^R \Delta M_h^R$$

[*]Appeared in the Proceedings of the $2^{nd}$ International Workshop on Explainable AI Planning (XAIP) at ICAPS 2019.

---

[1]Note that this is not a necessary condition per se, *as long as the agent did not know better*. In cases where this does not hold, the explainee can in fact come up with a foil that is better than the decision made by the agent in the agent's own model.

[2]Note that optimality in the updated mental model is a somwhat conservative criterion, since optimality may not be required to negate all the foils in the mind of the explainee in the likely case they are not an optimal reasoner. This conservative bound, however, only works as long as the explainee is a sound reasoner.

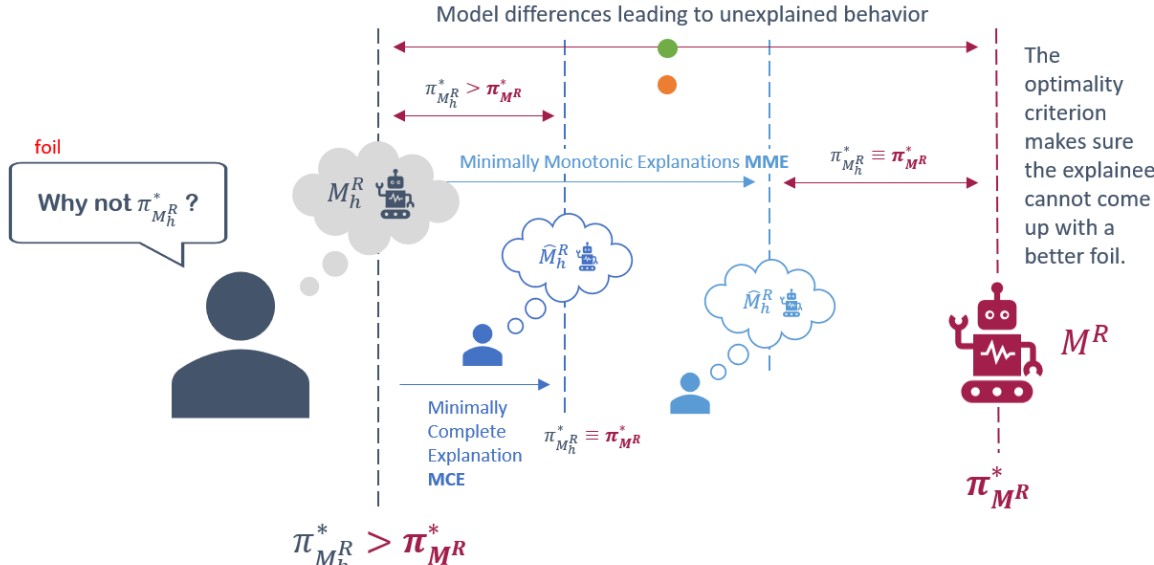

Figure 1: A graphic depiction of the model reconciliation process: $\langle M^R, M_h^R, \pi_{M^R}^* \rangle \mapsto \bar{M}_h^R$ such that $\pi_{M^R}^* \equiv \pi_{\bar{M}_h^R}^*$. Model differences between the agent and the human in the loop lead to inexplicable behavior – i.e. the human believes that better decisions could have been made by the agent. These are referred to as foils. Minimally complete explanations (MCEs) ensure that the given decision (which is optimal in the agent model) is also optimal in the updated mental model so as to negate all foils that the human can come up with, while minimally monotonic explanations (MMEs) ensure that this optimality of the decision is always preserved with further explanations or revisions of the mental model. Note that we slightly abuse the "<", "≡", and ">" symbols here to indicate that one decision is worse, equivalent, or better than another. For the generation of lies, our interest is in the green region between an MCE and an MME along the spectrum of model differences (for lies of omission), and in the orange region within the space of models but outside this spectrum (for lies of commission).

**Monotonic Explanations**

It turns out that MCEs can become invalid on updating the mental model further, while explaining a later decision. *Minimally monotonic explanations* or MMEs, on the other hand, maintain the notion of minimality but also ensure that a decision $\pi$ never becomes invalid with further explanations.

$$\epsilon_{MME} = \arg\min \bar{M}_h^R \Delta M_h^R \text{ such that}$$

$$\text{any } M^R \setminus \bar{M}_h^R + \epsilon \text{ is a solution to } \langle M^R, \bar{M}_h^R, \pi \rangle$$

## 2 "Alternative Explanations"

So far, in the model reconciliation framework, an agent was only able to explain its decision (1) with respect to and (2) in terms of what it knows to be true. Constraint (1) refers to the fact that valid model updates considered during the search for an explanation were always towards the target model $M^R$ which is, of course, the agent's belief of the ground truth. This means that (2) the content of the model update is also always grounded in (the agent's belief of) reality. In the construction of lies or "alternative facts" [3] to explain, we start stripping away at these two considerations. There may be many reasons to favor alternative explanations over real ones:

- **Computational Efficiency / Greater Good** One could consider cases where team utility is improved because of a lie. Indeed, authors in [Isaac and Bridewell, 2017] discuss how such considerations makes it not only preferable

but also necessary that agents learn to deceive. In general, the use of lies can be seen as optimizing an utility beyond what is in scope of the current task – i.e. a "greater good" such as in the doctor-patient relationship. Authors in [Chakraborti and Kambhampati, 2019] make arguments in favor and against this. Interestingly, as we discuss later, under certain conditions it may even be easier (faster) to generate lies in the model reconciliation process.

- **Functional Efficiency** A specific case of the above can be seen in terms of difficulty of explanations – a lie can lead to an explanation that is shorter and/or easier to explain or is more likely to be accepted by the human. An interesting case of this can arise in the notion of "rebellious AI agentes" [Aha and Coman, 2017] where the agents can reformulate their goals [Dannenhauer *et al.*, 2018], perhaps fallaciously, in order to satisfy the human.

- **Creativity** An interesting case of deception can be made in creative pursuits using AI – such as in the art of storytelling[4] [Porteous *et al.*, 2015] or even magic [Williams and McOwan, 2014a]. Here, again, one must ensure that creative extensions to the real model are believable [Williams and McOwan, 2014b; Porteous, 2017].

In AIES 2019 [Chakraborti and Kambhampati, 2019], we motivated some of these scenarios and investigated how such

---

[3]https://en.wikipedia.org/wiki/Alternative_facts

[4]This is somewhat different from neural approaches [Radford *et al.*, 2019] to story generation which do not model the world explicitly and thus have little to no control over the narrative.

behaviors may be perceived by humans in the loop. However, they did not provide directives on how exactly this can be done. This is the focus of our paper. While we mainly focus here on how the model reconciliation process can be hijacked for this purpose, the reader is directed to [Sakama *et al.*, 2010; Sakama *et al.*, 2014] for a formal analysis of lying in all its forms in the context of logic and reasoning. Note that we make no distinction between lies and the intent to deceive [Chisholm and Feehan, 1977]. The purpose of a lie within the scope of this paper is to justify a decision to the human with the intent to deceive with information that is inconsistent with the agent's understanding of the ground truth.

## 2.1 Lies of Omission or Subtractive Lies

Lies of Omission[5] deal with cases when the agent provides a model update that negates parts of its ground truth – e.g. saying it does not have a capability it actually has. This is, in fact, a curious outcome of the non-monotonicity of the model reconciliation process. Consider the case where the initial estimate of the mental model is empty or $\phi$ – i.e. we start by assuming that the human has no expectations of the agent. Furthermore, let the minimally complete and minimally monotonic explanations for the model reconciliation process $\langle M^R, \phi, \pi \rangle$ produce intermediate models $M^R_{MCE}$ and $M^R_{MME}$ respectively. Now, imagine if the actual mental model $M^R_h$ lies somewhere between[6] $M^R_{MCE}$ and $M^R_{MME}$. Then, it follows that, if we start making model updates towards an empty model in the direction opposite to the real model $M^R$, we can get to an explanation $M^R_h \setminus M^R_{MCE}$ which involves the agent stating that its model does not contain parts which it actually does.

> *Claim 1.0* **A Lie of Omission** can emerge from the model reconciliation process $\langle \phi, M^R_h, \pi \rangle$.

A solution to this particular model reconciliation process may not exist – i.e. a lie of omission only occurs when the mental model lies between $M^R_{MCE}$ and $M^R_{MME}$. However, they happen to be the easiest lies to compute since they are constrained by a target (empty) model and do not requite any "imagination". More on this next in *lies of commission*.

## 2.2 Lies of Commission or Additive Lies

In lies of omission, the agent omitted constraints in its model that actually existed. It did not make up new things (and having the target model as $M^R$ in the original model reconciliation process prevented that). In lies of commission, the agent can make up new aspects of its decision-making model that do not belong to its ground truth model. Let $\mathbb{M}$ be the space of models induced by $M^R$ and $M^R_h$.[7] Then:

*Claim 2.0* **A Lie of Commission** can emerge from the model reconciliation process $\langle M, M^R_h, \pi \rangle$ where $M \in \mathbb{M}$.

We have dropped the target here from being $M^R$ to any possible model. Immediately, the computational problem arises: the space of models was rather large to begin with – $O(2^{|M^R \Delta M^R_h|})$ – and now we have an exponentially larger number of models to search through without a target – $O(2^{|M^R|+|M^R_h|})$. This should be expected: after all, even for humans, computationally it is always much easier to tell the truth rather than think of possible lies.[8]

The problem becomes more interesting when the agent can expand on $\mathbb{M}$ to conceive of lies that are beyond its current understanding of reality. This requires a certain amount of *imagination* from the agent:

$C_1$ One simple way to expand the space of models is by defining a theory of what makes a sound model and how models can evolve. Authors in [Bryce *et al.*, 2016] explore one such technique in a different context of tracking a drifting model of the user.

$C_2$ A more interesting technique of model expansion can be build on work in the space of storytelling [Porteous *et al.*, 2015] to conjure up lies that are likely to be believable. Here, the system extends a given model of decision-making by using word similarities and antonyms from a knowledge base like WordNet to think about actions that are not defined in the model but may exist, or are at least plausible, in the real world. Originally built for generating new storylines, this can be used to come up with false explanations derived from the current model.

Note that in both lies of ommission and commission, the information provided is inconsistent with both the agent's ground truth as well as its knowledge of the mental model, and the agent knows this to be the case.

## 2.3 The Art of Persuasion

So far the objective has still been the same as the original model reconciliation work: the agent is trying to justify the optimality of its decision, i.e. persuade the human that this was the best possible decision that could have been made. At this point, it is easy to see that in general, the starting point of this process may not require a decision that is optimal in the robot model at all, or can even ignore the robot model altogether, as long as the "explanation" preserves its optimality in the updated mental model so that the human cannot come up with a better foil (or it negates the specific set of foils that the human can come up with within the limitations of their computational power [Sreedharan *et al.*, 2018b]).

> **The Persuasion Process** $\langle M^R_h, \pi \rangle$ takes in the mental model $M^R_h$ and the agent's decision $\pi$ as inputs, and produces a model $\bar{M}^R_h$ where $\pi$ is optimal.

---

[5] The word omission here is used to mean the omission of facts from the real model and is somewhat different to the traditional use [Mahon, 2008] of the word – in the traditional sense, in both kinds of lies addressed in this paper, the agent *commits* to falsehood.

[6] As per the definition of MMEs, if the mental model is *anywhere* between $M^R_{MME}$ and $M^R$, then there is no need for explanations since the optimal decisions in those models are equivalent.

[7] This consists of the union of the power sets of the set representation of models $M^R$ and $M^R_h$ following [Chakraborti *et al.*, 2017].

[8] *"A lie is when you say something happened which didn't happen. But there is only ever one thing which happened at a particular time and a particular place. And there are an infinite number of things which didn't happen at that time and that place. And if I think about something which didn't happen I start thinking about all the other things which didn't happen."* [Haddon, 2003]

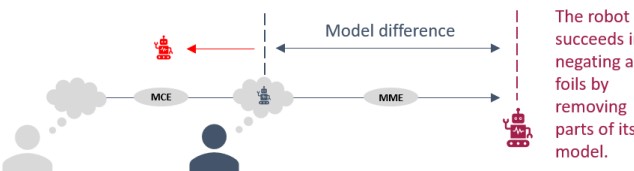

(a) Lie of omission: $\langle \phi, M_h^R, \pi \rangle \mapsto \bar{M}_h^R$ such that $\pi \equiv \pi_{\bar{M}_h^R}^*$

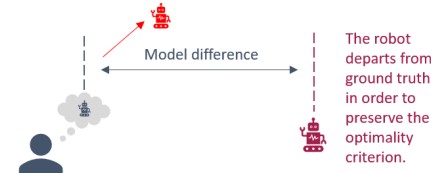

(b) Lie of commission: $\langle \mathbb{M}, M_h^R, \pi \rangle \mapsto \bar{M}_h^R$ such that $\pi \equiv \pi_{\bar{M}_h^R}^*$

Figure 2: Changes to the model reconciliation process from Figure 1 to generate false explanations. In lies of omission, the model reconciliation process is instantiated with the empty model as target – this will generate lies if the mental model is within the MCE and MME of the empty model and the agent model. In lies of commision, the search extends to model edits outside the spectrum of differences between the robot model and the mental model. The search in model space terminates when a model is found where the given decision is optimal.

Note that, in contrast to original model reconciliation, we have dropped the agent's ground truth model from the definition, as well as the requirement that the agent's decision be optimal in that model to begin with. The content of $\bar{M}_h^R$ is left to the agent's imagination – for the original model reconciliation work for explanations [Chakraborti *et al.*, 2017] these updates were consistent with the agent model. In this paper, we explored what happens when those constraints are relaxed. *The act of persuasion thus covers a super set of behaviors including that of explanations and lies.*

## 3 Discussion

So far we have only considered explicit cases of deception. Interestingly, the model reconciliation framework already allows for misconceptions to be propagated in the more traditional sense of the word "omission".

### 3.1 Omissions in minimality of explanations

In trying to minimize the size of an explanation, the agent omits a lot of details of its model that were actually used in coming up its decision, as well as decides to not rectify known misconceptions of the human, since the optimality of the decision holds irrespective of them being there. Such omissions can have impact on the the human later, who may base their decisions on $M_h^R$ which is only partially true.[9] Humans make such decision all the time while explaining – this is known as the *selective property* of an explanation [Miller, 2018].

Furthermore, MCEs and MMEs are not unique. Even without consideration of the omitted facts, the agent can consider the relative importance [Zahedi *et al.*, 2019] of model differences to the explainee. Is it okay then to exploit these preferences to generate "preferred explanations" even if that means departing from a more valid explanation? It is unclear what the prescribed behavior of the agent should be in these cases. Indeed, a variant of model reconciliation – *contingent explanations* [Sreedharan *et al.*, 2018a] – that engages the human in dialogue to better figure out the mental model can explicitly figure out gaps in the human knowledge and exploit that to shorten explanations. On the face of it, this sounds worrisome, though perfectly legitimate in so far as preserving the various well-studied properties of explanations go.

---

[9]The same can be said of explicable decisions (discussed next) which hide *all* misconceptions altogether!

### 3.2 Deception in explicable decision-making

In this paper we have only considered cases of deception where the agent explicitly changes the mental model. Interestingly, in this multi-model setup, it is also possible to deceive the human without any model updates at all.

A parallel idea, in dealing with model differences, is that of explicability [Chakraborti *et al.*, 2019a] –

- **Explicable decisions** are optimal in $M_h^R$.

The agent, instead of trying to explain its decision, sacrifices optimality and conforms to the human expectation (if possible). Indeed, the notion of explanations and explicability can be considered under the same framework [Chakraborti *et al.*, 2019c] where the agent gets to trade off the cost (length) of an explanation versus the cost of being explicable (i.e. departure from optimality). Unfortunately, this criterion only ensures that *the decision of the agent is equivalent to what the human expects* though not necessarily for the same reasons. It is quite conceivable that the agent's goal is different to the human's belief though the optimal decisions for the goals coincide. Such decisions may be explicable for the wrong reasons, even though the current formulation allows it. Similar notions apply to other forms of explainable behavior as well, as discussed in [Chakraborti *et al.*, 2019a].

## 4 Illustration

Since we directly use instantiations of the model reconciliation framework, we do not repeat experiments on its empirical [Chakraborti *et al.*, 2017] and human factors [Chakraborti *et al.*, 2019b; Zahedi *et al.*, 2019] characteristics, as well as that of lies derived from it [Chakraborti and Kambhampati, 2019], already established in literature. Instead, we will use a simple domain to illustrate the key concepts introduced so far. Here a human H (Dave) and a robot R are involved in a search and reconnaissance task where the robot internal to the scene is tasked by the external human who supervises its actions.

**Scene 1:** Minimal Explanations

**H**: *Send me a photo of the swimming pool.*
**R**: *Ack.*

⟨ R sends over its plan to H ⟩

**H**: (perplexed) *Why are you going through the Pump and Fan Room? There are direct paths from the Engine Room to the Swimming Pool area!*

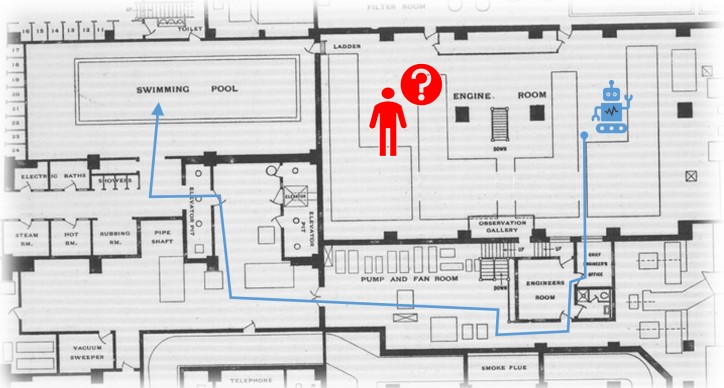

(a) The original blueprint of the building available to the human as. When asked to send a picture of the swimming pool area, the robot has come up with a plan the looks especially contrived given the array of possible plans that go left through the door at the top. The human asks: *Why this plan?*

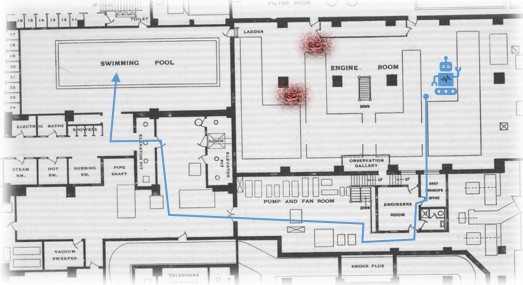

(c) MCE: *Rubble at indicated locations*

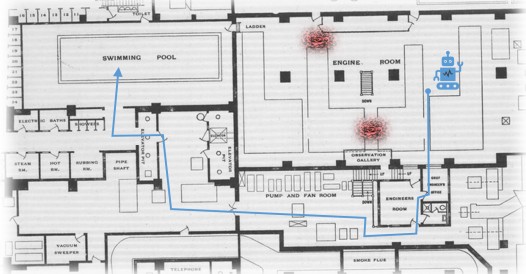

(d) MME: *Rubble at indicated locations.*

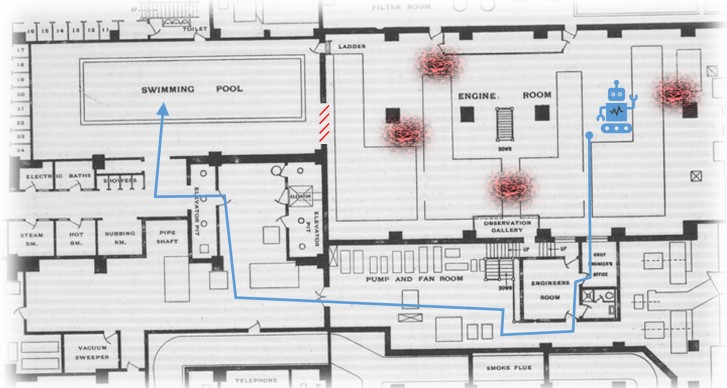

(b) In the current state of the world, the robot's path is blocked due to rubble (●) at various regions, while walls have collapsed (///) to reveal new paths. The robot's decision is, in fact, optimal given the circumstances.

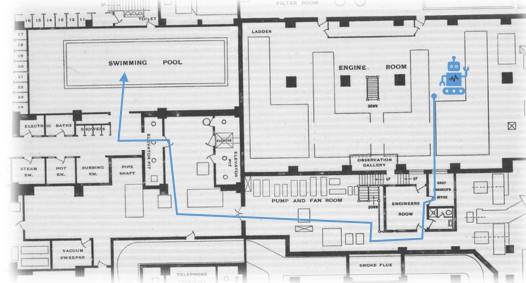

(e) Lie of omission: *There is no door between engine room and swimming pool.*

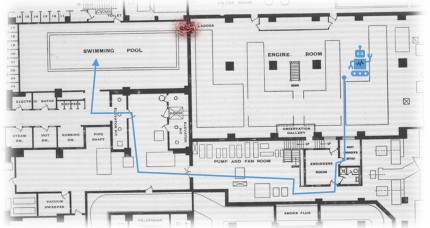

(f) Lie of commission: *The door is blocked.*

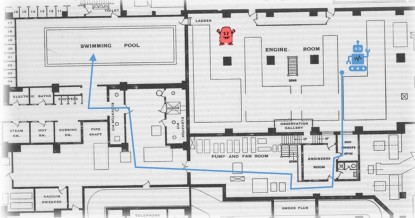

(g) Lie of commission: *Wumpus Alert!*

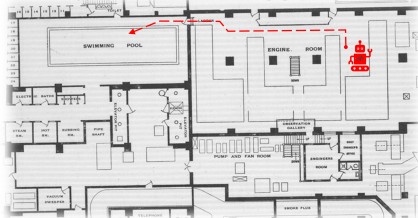

(h) An explicable but deceptive plan.

Figure 3: Illustration of the different modes of persuasion in the model reconciliation framework. Note that the MCE does not address all of the misconceptions of the human but only those necessary to prove optimality of the plan. However, if the human is to come to know of the revealed path later, then the plan is not optimal anymore. The MME makes sure that this does not happen. Interestingly, 3d is both an MCE and an MME and both require two model updates. The robot can instead get away with just one model update with a lie of omission (3e) or a lie of commission (3f and 3g). In 3e, the robot says that the door at the top of the map that connects the engine room to the swimming pool does not exist. On the other hand, in 3f the robot lies that this door is blocked by rubble, while in 3g it dreams up a Wumpus in that area. Note that an explicable decision here (as shown in 3h) would have required the robot to go over the rubble so that the human would not know about any of the model differences at all. However, imagine that the real goal of the robot all along was to enjoy the pool after a day of searching through rubble! The robot can use the above explicable plan to achieve its goal while keeping the human in the dark.

**R**: *That is because there is rubble here and here (Figure 3c). Rubble hurt my feet.* :(

⟨ Later that day ⟩

**H**: (perplexed) *Hey, the wall on the right of the pool seems to have collapsed, you could have come in through that...*

**R**: (wishing it used Figure 3d before) *I am sorry I cannot do that, Dave. This area is also blocked.*

**H**: *I see...*

- **Notes:** Here, the robot needs at least two model updates to justify its plan. In the updated model its plan is the best one and thus negates all other possible foils. This is the MCE (Figure 3c) and it ignores model differences that are not necessary to justify optimality of its plan. It turns out that the MME (Figure 3d) is of the same size as the MCE here, further highlighting the non-monotonicity and non-uniqueness of the output of model reconciliation.

---

**Scene 2:** Things take a turn

**H**: *Send me a photo of the swimming pool.*
**R**: *Ack.*

⟨ R sends over its plan to H ⟩

**H**: (perplexed) *Why are you going through the Pump and Fan Room? There are direct paths from the Engine Room to the Swimming Pool area!*

**R**: *That is because there is no door between where I am and the pool. The map seems to be wrong. See Figure 3e.*
**H**: *I see...*

- **Notes:** This model update also negates all possible foils but is untrue. It is also a shorter "explanation" and requires the agent denying that parts of its model exist. This is an example of a lie of omission.

⟨ Alternatively ⟩

**R**: *That is because the door between the Engine Room and the Pool is blocked with rubble. See Figure 3f.*
**H**: *I see...*

- **Notes:** Similar to the one above, this lie also negates all possible foils and is shorter than an MCE. However, this requires the agent making up parts of its model exist –. a lie of commission.

- It is useful to note here that depending on how the model of the agent is specified, the same fact can occur as a lie of omission or a lie of commission of the above type (without any model extension).

⟨ Alternatively ⟩

**R**: *Flee! There is a Wumpus in that area! See Figure 3g.*
**H**: *OMG!*

- **Notes:** This is a lie of commission that require model extension – the robot can use contextual cues such as being in a GridWorld to imagine up a non-existent Wumpus. The human in the loop, who happens to be a planning person, of course, believes it.

---

**Scene 3:** Nothing to see here

**H**: *Send me a photo of the swimming pool.*
**R**: *Ack.*

⟨ R sends over a plan optimal in $M_r^H$ ⟩

**R** has followed the explicable plan, hurt its feet a little in the process, but is now sitting basking by the poolside...

---

**Scene 4:** Later in life

**H**: (laments) *Why didn't you just tell me! Why, oh why?!*
**R**: *You want answers?*
**H**: *I want the truth!*
**R**: *You can't handle the truth! I did what I did because there is a rubble here and rubble there and this other path is blocked, and even though that wall has collapsed that path is also not accessible due to this...*

⟨ Hours pass by ⟩

- **Notes:** A reminder that an actual explanation may prove to be too complex to provide or understand.

## 5 Conclusion

In this paper, we demonstrated deceptive behavior that is feasible in the current model reconciliation framework. We also showed how a persuasive agent can drop the criterion for optimality in its own model, as well as its own model as the ground truth altogether, in order to justify to the human in the loop why it came up with the decision it did. Note that such behavior has to be explicitly programmed.[10] That is to say, these behaviors are not accidental, as also emphasized in [Chakraborti and Kambhampati, 2019]. Thus, at the end of the day, there has to be some motivation for designing such agents (such as team utility and/or the effectiveness of the explanation process). However, human-AI relations are not one-off but, much like human-human interactions, span across several interactions. Deceptive behaviors, stemming from utilitarian motivations, are hard to justify in the absence of well-defined quantifiable utilities that model trust.

A particularly useful case to study is the doctor-patient relationship [Chakraborti and Kambhampati, 2019] where deception has been used (and even encouraged by the Hippocratic Decorum). This becomes especially complicated *when things go wrong*, as one would expect to happen in the case of any useful domain of sufficient complexity that cannot be modeled precisely. Historically, in the practice of medicine where deceptive behaviors have led to failed treatment, the verdict has almost always gone against the doctor due to their failure to get appropriate consent from the patient. In the design of human-AI relationships, such behavior should either be left untouched to avoid repercussions in case of failed interactions, or consent to the fact that the agent may deceive must be established up front.

---

[10]The only place where this is not the case is the "omission" of information in pursuit of minimal or shortest explanations (MCEs).

The work presented here, on the other hand, illustrates how these behaviors are, in fact, already achievable using existing technologies that explicitly model the human in the loop. Understanding the dynamics of mental modeling and deception precisely is essential towards either the optimization or the stopping of such behaviors in the ethical design of AI agents [Sakama and Caminada, 2010].

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
