# OpenReview forum: "(How) Can AI Bots Lie?"
_icaps-conference.org/ICAPS/2019/Workshop/XAIP — XAIP 2019_

### Official Review · AnonReviewer2 · 2019-05-01
**A planning-centred view of deception in explanation**

**Rating:** 3
**Confidence:** 2

**Review:**

This paper presents an initial look at deception in explanation and explicibility. It uses an existing framework on explanation as model reconciliation to define lies of omission and lies of commission.

The model presented seems sensible and as far as I can tell is consistent with existing definitions of lies of omission/commission. The example is very nice and entertaining; I love the implicit reference to '2001 A Space Odyssey' :)  As the paper is preliminary, there is no prescreptive way to generate lies and therefore no evaluation/results.

A few minor comments:

- To be honest I found the framing of the paper quite strange. The abstract and 'Alternative Explanations' section are written as though the reader should find it surprising that model reconciliation process can be used for deception. Given that model reconiliation is an instance of a model of theory of mind, this is really quite clear -- the relationship between theory of mind and deception is well established both within and outside computer science. It would be more curious if a specific instance (model reconciliation) could not be used for deception!

- Regarding the subtitle "A Formal Perspective on the Art of Persuasion", I think calling it 'Persuasive Deception' is more accurate. Persuation need not be deceptive in general.

- It is not clear to me what footnote 1 means. Nor does I think it is necessary. Even though I don't understand it, I don't agree! There are many causes for the need of explanation.

- I didn't quite understand the first part of the "Lie of omission" section. Negating part of the ground truth sounds a like like commission to me, rather than omitting part of the ground truth?

- The references are all quite 'planning' focussed. Given that this is a planning workshop this is not surprusing, but g, there is a very large body of work around on deception in AI that is relevant, even though the focus is on explanation/explicibility in this paper (especially given that the subtitle is "A Formal Perspective on the Art of Persuasion"). In particular, work from Chiaki Sakama; e.g.: https://hal.archives-ouvertes.fr/hal-01387822/document which also defines lies of omission/commission, but Sakama has more work and builds strongly on foundations that could be exploited as the work in the current paper matures. As it currently stands, the definitions in the paper are somewhat 'ad-hoc'.

- Another reference that would be highly interesting for the authors is "The Intent to Deceive" by Chisolm and Feeham: The Journal of Philosophy, Volume 74, Issue 3, March 1977

- At the end of the paper, the authors note that one must be careful in the use of this, but there are several examples of situations in which deception is absolutely a good thing: the most obvious being doing magic.

Minor typos:

- "there is a rubble here and another rubble" --> "there is rubble here and rubble there" or "there is a pile of rubble here and a pile of rubble there"

---

> ### Author Response · Authors · 2019-05-15
> **thank you for the review**
>
> Thanks a lot for the thoughtful comments.
>
> > The example is very nice and entertaining; I love the implicit reference to '2001 A Space Odyssey' :)
>
> Did you miss the reference to A Few Good Men! :P
>
> > As the paper is preliminary, there is no prescreptive way to generate lies and therefore no evaluation/results.
>
> Actually, the two instantiations of the model reconciliation process on page 2 are prescriptive, lies of omission can be computed with the existing code if instantiated that way while the lies of commission would require little changes. But yes, I agree we did not evaluate it. Partly because I am still looking for a compelling use case where lies make complete sense and we could actually run studies on the impact of those lies, rather than an evaluation of the computational aspects of it which are expected to be similar to explanations in the model reconciliation framework. If you have any suggestions for an example domain that would be great too. Magic is a great example!
>
> > It would be more curious if a specific instance (model reconciliation) could not be used for deception!
>
> Point taken :) I was more surprised that the original framework already allows for that without any extensions, but I probably shouldn't have extended that.
>
> > Regarding the subtitle "A Formal Perspective on the Art of Persuasion", I think calling it 'Persuasive Deception' is more accurate. Persuation need not be deceptive in general.
>
> Hmm, that is true. That is why in the definition of the persuasion process, there is no mention of deception. Explanations also follow that. We would make this clear.
>
> > It is not clear to me what footnote 1 means. Nor does I think it is necessary. Even though I don't understand it, I don't agree! There are many causes for the need of explanation.
>
> If the underlying model and the computation on it are same, I struggle to think what the cause of an explanation would be? Since then the human would come to the same conclusions as the planner. Maybe we disagree on what the model has within it?
>
> > I didn't quite understand the first part of the "Lie of omission" section. Negating part of the ground truth sounds a like like commission to me, rather than omitting part of the ground truth?
>
> I have also struggled with the same question but haven't arrived at a satisfactory alternative yet. My reasoning was that at least in the case of omissions, you are not making new stuff up. I will find a better name.
>
> - references
>
> Thanks so much for these! I was only aware of the "white lies on silver tongues" paper in the literature and I was looking for more on what has been done in this space. The references will help a lot, we will fold them into the revised version.

---

### Official Review · AnonReviewer1 · 2019-05-09
**Intriguing paper with detailed examples**

**Rating:** 3
**Confidence:** 2

**Review:**

This paper is well written and motivated. The paper presents an approach that details how the model reconciliation process can be used for deception. This includes explanations that are in fact, lies. The paper considers lies of omission and commission, as well as multiple example walkthroughs. Highly relevant to the workshop and should spark interesting discussion.

This paper raises a number of very interesting questions for human-robot interaction:

Would humans prefer a lie rather than the truth if it's simpler? Perhaps some of the best explanations , under certain circumstances, could be lies?

In the conclusion the paper states that most of the deceptive behavior described in the paper needs to be explicitly programmed. I wonder if a more developed agent capable of rebelling might lie to enforce ethical principles, perhaps in the case the goal asked of the agent is one of ill-intent. In such circumstances, the rebellion process might inform the deceptive behavior without it having to be explicitly programmed.

---

> ### Author Response · Authors · 2019-05-12
> **rebellion**
>
> That is an interesting perspective! :) Indeed this can be an alternative motivation for resorting to persuasion using lies (other than our motivations of shorter or easy to compute/communicate "explanations"). We will add a discussion to this effect in the paper.

---

> ### Author Response · Authors · 2019-05-21
> **rebellion reference**
>
> What would be the definitive reference for planning and rebellious agents? This: https://www.aaai.org/ojs/index.php/aimagazine/article/view/2762?

---

### Decision · Program_Chairs · 2019-05-15

**Decision:**

Accept

**Comment:**

The reviewers agree to accept. Please address all review criticism as best possible for the final paper version and its presentation at the workshop. Looking forward to discuss your work at the workshop!